# RETRACTED: The *Dosidicus gigas* Collagen for Scaffold Preparation and Cell Cultivation: Mechanical and Physicochemical Properties, Morphology, Composition and Cell Viability

**DOI:** 10.3390/polym15051220

**Published:** 2023-02-28

**Authors:** Veronika Anohova, Lyudmila Asyakina, Olga Babich, Olga Dikaya, Aleksandr Goikhman, Ksenia Maksimova, Margarita Grechkina, Maxim Korobenkov, Diana Burkova, Aleksandr Barannikov, Anton Narikovich, Evgeny Chupakhin, Anatoly Snigirev, Sergey Antipov

**Affiliations:** 1Immanuel Kant Baltic Federal University, Nevskogo 14, Kaliningrad 236006, Russia; anohovaveronika@yandex.ru (V.A.); alk_kem@mail.ru (L.A.); olich.43@mail.ru (O.B.); ODikaya@kantiana.ru (O.D.); AGoikhman@kantiana.ru (A.G.); KMaksimova@kantiana.ru (K.M.); korobenkovmv@gmail.com (M.K.); abarannikov95@gamil.com (A.B.); narikovich@gmail.com (A.N.); chupakhinevgen@gmail.com (E.C.); anatoly.snigirev@gmail.com (A.S.); 2Voronezh State University, 1, University Square, Voronezh 394063, Russia; grechkina@phys.vsu.ru (M.G.); diana.burkova488@gmail.com (D.B.); 3Saint Petersburg State University, Saint Petersburg 199034, Russia

**Keywords:** collagen scaffolds, mechanical properties, physicochemical properties, morphology, composition, cell viability, molecular structure, *Dosidicus gigas*, X-ray tomography, SynchrotronLike

## Abstract

Directed formation of the structure of the culture of living cells is the most important task of tissue engineering. New materials for 3D scaffolds of living tissue are critical for the mass adoption of regenerative medicine protocols. In this manuscript, we demonstrate the results of the molecular structure study of collagen from *Dosidicus gigas* and reveal the possibility of obtaining a thin membrane material. The collagen membrane is characterized by high flexibility and plasticity as well as mechanical strength. The technology of obtaining collagen scaffolds, as well as the results of studies of its mechanical properties, surface morphology, protein composition, and the process of cell proliferation on its surface, are shown in the given manuscript. The investigation of living tissue culture grown on the surface of a collagen scaffold by X-ray tomography on a synchrotron source made it possible to remodel the structure of the extracellular matrix. It was found that the scaffolds obtained from squid collagen are characterized by a high degree of fibril ordering and high surface roughness and provide efficient directed growth of the cell culture. The resulting material provides the formation of the extracellular matrix and is characterized by a short time to living tissue sorption.

## 1. Introduction

Collagen is the main protein of connective tissue, which consists of three subunits in the form of an alpha helix [1,2,3]. The size of the turns of the fibrillar structure of collagen is 67 nm (D-period). There are 20 types of collagen proteins; the main type is type I collagen (collagen I) [4]. Collagen proteins have the property of self-structuring in solution and in the process of drying and lyophilization [5], which makes it possible to obtain collagen materials, particularly matrices, which have an ordered structure and are similar to the connective tissues of the human body [6,7,8]. Self-assembly of collagen molecules into supramolecular structures—fibrils—is the main process of collagen matrix formation [9]. Collagen I is used in medicine for early healing, skin buildup [10], as suture material, and as artificial skin [11]. Implantable materials obtained from collagen type I do not induce an immunological rejection; such materials are highly biocompatible and capable of controlled degradation [12]. Collagen scaffold materials can be obtained from collagen I solutions by casting, lyophilization, electrospinning, and coagulation. The use of this technology makes it possible to obtain collagen materials with an adjustable fiber size, which is used to obtain substrates in cell technologies [13]. The formation of structures similar to native collagen is difficult for the technologies that are available today. Electrospinning and deposition technologies cannot establish the long-range orientation of collagen fibers [14]. Collagen films obtained by lyophilization are not able to copy the extracellular matrix of living biological systems [15,16]. Today, the investigation of tissue proliferation processes and cell differentiation is carried out by real-time X-ray tomography technology [17,18,19,20]. At the same time, investigations of scaffolds’ surface topography for tissue engineering based on collagens of various origins are known [21,22]. The results of investigations of collagen molecules’ surface topography have made it possible to estimate the value of the D-period of fibrils, which is 68 ± 0.8 nm for collagen proteins [23]. Parameters of X-rays generated by synchrotrons are of particular importance for the research fields of regenerative medicine and cell and tissue engineering, as well as for the investigation of atomic and electron environments of composite bioorganic macromolecules [24,25,26,27,28]. Previously, we studied the molecular structure features of collagen from *Dosidicus gigas* [29]. At the same time, there exist results giving evidence that materials based on collagen proteins of hydrobionts inhabiting the South Atlantic are highly durable and biocompatible [30,31]. We focused our attention on the study of the living cell proliferation process on the surface of scaffolds obtained from *Dosidicus gigas* collagens, because the high availability of raw materials for obtaining collagen will allow the introduction of tissue engineering technique as a generally available method of therapy. We have investigated the characteristics of collagen proteins, their molecular structure features, amino acid sequences of their peptides, the strength of the surface and prepared collagen matrix morphology, as well as the proliferation process with the identification of the tissue formation area using soft X-ray radiation generated by a synchrotron source.

## 2. Materials and Methods

Commercially available reagents, nutrient media, and cell cultures were used in this research according to standard protocol. Cell imaging reagents were from Thermo Fisher and used according to the manufacturer’s protocols.

### 2.1. Extraction of Collagen 

Extraction of acid-soluble collagen was carried out according to the method published in [32]. The skin of the hydrobiont *Dosidicus gigas* was separated from the mantle and lyophilized at −20 °C and a pressure of 0.03 mbar. Lyophilized skin was homogenized on an IKA mechanical disperser at 2600 rpm. The powder weighing 50 g was placed in a flat-bottomed flask. For the extraction of non-collagen proteins and lipids, 250 mL of PBS buffer solution, pH 8.4, containing 2000 U of trypsin and 2500 U phospholipase A was added. The mixture was sonicated (2.4 kHz, 200 W) for 1 min every hour at room temperature for 6 h. Collagen proteins were separated by centrifugation at 2500 rpm and transferred to a 500 mL flat-bottomed flask containing 300 mL of 0.2 M NaOH and mixed for 6 h. Next, collagen proteins were precipitated by centrifugation at 2500 rpm and the resulting precipitate was used to isolate acid-soluble collagen. To obtain 300 mL of acid-soluble collagen, a 3% solution of acetic acid was added to the precipitate and mixed for 3 h under periodic sonication of ultrasound with the same parameters as above. The insoluble fraction was separated by centrifugation at 1500 rpm. The resulting colloidal solution was neutralized by 0.2 M NaOH to pH 7.4, and the precipitate of type I collagen was isolated by centrifugation at 3500 rpm. The precipitate washed by three portions of distilled water until the acid reaction disappeared, and then the obtained sample was lyophilized.

### 2.2. Preparation Collagen Scaffold

The technology for obtaining collagen material was developed. Collagen weighing 10 g was suspended in 50 mL of a 3% glycerol solution in distilled water under ultrasound treatment. The suspension was placed on a Teflon substrate and air dried at 120 °C in a convection dryer chamber until the moisture content in the sample was less than 3% under constant conditions. Moisture content was controlled by the standard gravimetric method. The sample was dried to a constant weight and the mass fraction of moisture was calculated as the difference in the masses of the sample before and after drying. The measurements were carried out in two repeats. Additionally, the moisture content was monitored using near-infrared spectroscopy. After scaffold drying, it was sterilized in UV at a wavelength of 280 nm and a temperature of 40 °C. The resulting collagen sheets were used to obtain ready-to-use cell scaffolds.

### 2.3. Mechanical Properties of Collagen Scaffold Investigation

The mechanical properties of the collagen scaffolds were estimated according to [33]. Tensile properties of 3D collagen matrices were measured using a modified Minimat 2000 mini-materials tester (Rheometric Scientific, Inc. (https://www.ssgca.com/), Piscataway, NJ, USA) and by standard testing procedures for matrices of low mechanical strength. One end of each specimen was attached to a stepper motor-controlled linear actuator and the other end was attached to a load cell. For these experiments, a load cell designed with a sensitivity of 0.0003 N was utilized. Clamps offset the loading axis and allowed the matrix to be submerged longitudinally within a bath of PBS, pH 7.4, at approximately 37 °C. For most experiments, the extension speed was 10 mm/min and the strain rate was 38.5 percent/min.

### 2.4. Atomic Force Microscopy

The study of the topography of the fibrillar structure and surface of collagen matrices was carried out according to [34]. AFM imaging and indentation of collagen fibrils were performed using a Nanowizard AFM (JPK Instruments, Berlin, Germany) and SOLVER P47 (NT-MDT, Russia). All measurements were taken in air at room temperature. Aluminum-coated silicon AFM cantilevers (resonance frequency up to 150 kHz and nominal spring constant up to 4.5 N/m (NSC12 tip-C, MikroMasch, Tallinn, Estonia)) and gold-coated silicon AFM cantilevers (resonance frequency up to 150 kHz and nominal spring constant up to 6.1 N/m (NSG 03, NT-MDT, Moscow, Russia)) were used. These cantilevers were chosen to match the stiffness of collagen for optimizing the sensitivity and signal/noise ratio (SNR). For imaging collagen fibrils without damage, the relatively high stiffness of the cantilevers required the use of intermittent contact mode (also known as tapping mode). After taking a topographic image of the fibril, the force mapping mode (also known as force volume mode) was used to perform quasi-static indentations. The 64 × 64 indentation curves, consisting of 256 data points each, were taken on a square area. The area’s dimensions were 1.5 × 1.5 mm ± 0.5 mm, optimized in each experiment to balance approximately the number of data points taken on the fibril and on the substrate. The time for both the approach and retraction of the cantilevers was set to 0.2 s (5 Hz), with zero delay in between.

### 2.5. Scanning Electron Microscopy

Collagen scaffolds were fixed overnight at 4 °C with 3% glutaraldehyde in 0.05 M phosphate buffer (PB) at pH 7.2. Following extensive rinsing with the same buffer at 4 °C, blocks were soaked for 1 h in 0.5% tannic acid in PB at 4 °C. Then, they were rinsed four times in the same buffer for 15 min at 4 °C and post-fixed with 1% osmium tetroxide in PB for 1 h at 4 °C. Samples were washed in distilled water, dehydrated in a graded series of ethanol, and freeze-dried in a Balzers Union CPD 020 (Balzers, Liechtenstein) using the procedure of critical point drying. Images of collagen scaffolds were obtained with scanning electron microscopy (SEM) in the bright field mode by Jeol JSM-6390 (Japan) with an accelerating voltage of 5 kV (low vacuum mode). As SEM demands a conductive surface to obtain an image, samples were covered with a platinum thin film (calibrated 30 nm) sputtered by the magnetron set-up Jeol JFC-1600 (Japan). 

### 2.6. Amino Acid Content by HPLC

For the preparation of dabsyl-cloride derivatized amino acids (dabsyl-AA), each amino acid mixture was dissolved in 0.1 N HCl solution . Dabsyl-Cl (26 mg) was dissolved in 20 mL of acetonitrile. A 0.1 M NaHCO_3_ solution (pH 8.3) and a 0.05 M Na_2_HPO_4_ solution (pH 7.0) were separately mixed with an equal volume of ethanol. A 0.5 mL aliquot of amino acid solution was placed in a test tube and mixed well with 0.5 mL of NaHCO_3_ solution and 2 mL of dabsyl-Cl solution. The test tube was incubated in a 70 °C water bath for 15 min before 3 mL Na_2_HPO_4_/ethanol solution was added to stop the reaction. Dabsyl-AA mixtures were resolved with a LiChroCART 125-2 Superspher 100 RP-18 column (Merck) end detection on a UV/VIS detector. The sample size was 10 µL. The flow rate was 0.5 mL/min. Both solvent A (H_2_O) and solvent B (acetonitrile) contained 0.1% (*v*/*v*) formic acid. The gradient was programmed as follows: 15% B for 3 min, linear gradient from 15% to 85% B for 60 min, and 85% B for 7 min. The relationship of the responding peak area and the concentration of each dabsyl-AA was calculated with Excel software (Microsoft Co., Seattle, WA, USA). Next, 100 mg of the collagen matrix was hydrolyzed in 6 N HCl and sealed in hydrolysis tubes under nitrogen atmosphere and incubated in an oil bath at 108 °C for 24 h. For the hydrolysate of the collagen matrix used for dabsyl-AA derivates, the preparation was as described above, and amino acid content was determined on HPLC with a UV/Vis detector.

### 2.7. Size Exclusion Chromatography

Size exclusion chromatography was performed by following [35] on chromatography system (Biorad, NGC) equipped with a fraction collector and a photometric detector. Fractionation was carried out on the column (ENRICH SEC 650 10 × 300 mm, BioRad) using 25 mM of acetate buffer as an eluant and a flow rate of 1 mL/min. Detection was carried out at a wavelength of 210 nm. To the column was applied 5 mL of a collagen colloidal solution in 25 mM of acetate buffer by means of a constant flow pump to a site exclusion chromatography column, followed by fractionation.

### 2.8. SDS-PAGE

Electrophoresis under a denaturing condition in the presence of sodium dodecyl sulfate (SDS-PAGE) was carried out according to the modified Laemmli method [36] in the presence of 8% polyacrylamide. Powdered samples of collagen (5 mg) were dissolved in 1 mL of 0.02 M sodium phosphate buffer (pH 7.2) containing 1% (*w*/*v*) SDS and 3.5 M of freshly prepared urea at 1 mg/mL concentration. The samples were mixed gently at 4 °C for 2 h to dissolve collagen and placed on PAGE. Protein markers produced by Thermo Scientific in the range of 10–200 kDa (product number 26614) and 20–120 kDa (product number 26612) were used as standards.

### 2.9. FTIR Spectra Registration

FTIR spectra were obtained using an FT-IR ATR Alpha spectrometer equipped with an attenuated total reflectance unit (Bruker, Germany) and an InGAs detector. For FTIR spectra acquisition, the system was used in point mode (aperture of 250 × 250 μm) with a 2.0 cm^−1^ resolution and using 30 scans in transmittance mode (n = 35 spectra), and spectra acquisitions were performed under complete N2 purge of the analytical system.

### 2.10. MALDI-TOF

All mass spectra data were acquired by the MALDI-TOF mass spectrometer Autoflex (Bruker Daltonic, Germany). MALDI-TOF settings were as follows: high-voltage output on ion source - 19.5 and 18.45 kV; high voltage output on first reflector - 8 kV; frequency of Nd:YAG laser - 1 GHz (500 shots). The process of lyophilization of collagen was carried out in a freeze dryer (Labconco Triad) in standard mode.

### 2.11. Cell Culture

Cell cultivation was carried out in parallel with [37,38]. The collagen matrix was placed into the wells of an 18-well plate, and 2 mL of DMEM culture medium containing glutamine, ceftriaxone, and penicillin was added. A suspension of HEK cells in 1 mL of DMEM containing 500 units of cells was added to each well and incubated for 4 days in an incubator at 37 °C and 5% CO_2_ partial pressure. The collagen matrix was removed, treated with appropriate dyes, and visualized using confocal microscopy.

### 2.12. MTT Assay for Toxic Effect Determination

The effects of the collagen matrix on cell viability were determined using the MTT colorimetric test. For the MTT test, collagen scaffold disks 30 mm in diameter and 60 µm thick were used. The MTT protocol was optimized to increase well volume from 200 µL to 2000 µL. The discs were cut from edge to center once to form a 5 mm high cone. The cone was placed on a culture plate with cells 60 mm in diameter. The culture medium containing the cells was added in a volume of 2 mL (so that the cones were completely submerged). Cultivation was carried out in an incubator in a CO_2_ atmosphere. The culture plates were placed on an orbital shaker to simulate the movement of the culture fluid. HEK cells were diluted with the growth medium containing 100 μM of CoCl_2_ for chemically induced hypoxia to 3.5 × 10^4^ cells per mL, and the aliquots (7 × 10^3^ cells per (200 μL) were placed in individual wells on 96-well multiplates (Eppendorf, Germany) and incubated for 24 h. The next day, the cells were treated with synthesized compounds separately at a concentration of 100 μM and incubated for 72 h at 37 °C in 5% CO_2_ atmosphere. Each compound was tested in triplicate. After incubation, the cells were then treated with 40 μL of MTT solution (3-(4,5-dimethylthiazol-2-yl)-2,5-diphenyltetrazolium bromide, 5 mg mL^−1^ in PBS) and incubated for 4 h. After an additional 4 h of incubation, the medium with MTT was removed and DMSO (150 μL) was added to dissolve the formazan crystals. The plates were shaken for 10 min. The optical density of each well was determined at 560 nm using the ClarioStar microplate reader (BMG Labteh, Germany). The tested collagen matrix was evaluated for the antiproliferative action in three separate experiments.

### 2.13. Fluorescent Dying

Cells cultured on a collagen scaffold were fixed with a 4% paraformaldehyde solution in PBS, washed with PBS + 0.5% Triton X-100, and washed with PBS. Cells were incubated with Mitotracker Red, DAPI, and FITC for 30 min according to the manufacturer’s protocol and washed with PBS. The obtained stained specimen was examined by confocal laser fluorescence microscopy.

### 2.14. Visualization of Extracellular Matrix

Unstained 3D matrices of type I collagen were polymerized on cover glasses and imaged using confocal refection microscopy as previously described [39], employing a direct Carl Zeiss LSM 800 confocal using either the 10× Plan-Neofluar (NA = 0.3), a 20× water immersion Plan-Apochromat (NA = 1.0) objective, or a 63× water immersion C-Apochromat (NA = 1.2).

### 2.15. Study of Cell Culture on the Surface of a Collagen Matrix at a Laboratory Complex, “SynchrotronLike”

The cells obtained on the surface of the collagen matrixes were fixed with a solution of the formalin and then studied by a high-resolution X-ray radiography technique on a scientific and educational multifunctional system for synchrotron studies called "SynchrotronLike" [40]. The main element of the system is the MetalJet D2+ 70 kV microfocus X-ray source from Excillum [41]. A liquid gallium–indium alloy with a characteristic line of 9.251 keV (Ga Kα1) is used as an anode in this source. The minimum size of the X-ray source is 5 µm, and the maximum power of the electron gun is 250 W. This results in a source brightness of 10^10^–10^11^ photons/sec/mm^2^/mrad^2^/(0.1%Δλ/λ), which is an order of magnitude higher than for conventional solid anode X-ray tubes and comparable with the brightness of a bending magnet of the synchrotron sources. A collagen scaffold with cells was placed on a motorized stage with five degrees of freedom (X, Y, Z, θ, ω) located at a 60 cm distance from the X-ray source. To record the images, the Rigaku X Sight Micron LC CCD camera installed 5 cm behind the sample was used. This camera was equipped with a lens with an optical magnification of 10×, providing a pixel size of 0.55 μm and 1.5 μm spatial resolution.

### 2.16. Image Processing of High-Resolution X-ray Radiography and Microscopy

ImageJ software was used to overlay images and determine the relative position of cells [42].

## 3. Results

### 3.1. Scaffolds’ Mechanical Properties, Surface Topography, and Composition

The scaffolds’ mechanical properties and surface topography were investigated using three independent techniques. It was found that the thickness of the obtained matrix according to Section 2.2 was 100–150 μm, the tensile strength was 80–100 MPa, the elongation ability was up to 47%, and Young’s modulus was 1.5 GPa (Figure 1(I)). 

Thus, the mechanical properties of the collagen matrix have evidenced the ability to use it as a potential model for cell scaffolds for surgery. The topography of the collagen matrix surface was studied by atomic force microscopy. The presence of a dense, self-structured fibrillar surface was observed. It was detected that the D-period of *Dosidicus gigas* collagen (Figure 1(II)) in the molecule’s helix is 65–69 nm, which corresponds to the parameter of the helix of most proteins of the mammalian and bird collagen family. Examination of the collagen scaffold surface by scanning electron microscopy made it possible to detect a developed pore structure, the cell size of which varies within 5–25 μm, which is an effective size of the cellular matrix for the regeneration of connective tissue in surgery, as well as a barrier matrix in transplantology [43,44,45] (Figure 1(III)), and in the next step of this investigation, these possibilities were tested.

According to the literature, collagen types I, III, IV, and V have several characteristic FTIR regions located in the ranges of 1.700–1.600 cm^–1^, 1.480–1.350 cm^–1^, 1.300–1.180 cm^–1^, and 1.100–1.005 cm^–1^, respectively [46]. In addition, there is evidence that FTIR spectra of fish skin collagens demonstrate the presence of amides (3423 and 3337 cm^−1^ for amide A and 2928 and 2924 cm^−1^ for amide B, respectively). In addition, bands correspond to stretching vibrations of C-O of the polypeptide backbone in the region of 1600–1700 cm^−1^ and hydrogen bonding between the N-H and C-O (Gly) residues at 1654 and 1560 cm^−1^. Signals for amide II and amide III in the structure of collagen were detected at 1240 cm^−1^ (48). At the same time, FTIR spectra obtained for collagen extracted from the skin of squid (*Doryteuthis singhalensis*) have similar characteristics [47]. 

FTIR investigations of the obtained scaffolds were carried out (Figure 2). The obtained data indicate the presence of bands in the regions of 3300 cm^−1^ and 2935 cm^−1^, which may correspond to amide A and amide B; a band in the region of 1659 cm^−1^, which can be correlated with the C-O backbone of the polypeptide chain; and a band in the region of 1538 cm^−1^, which is consistent with regions characteristic of N-H and C-N. A band was also detected in the region of 1235 cm^−1^, which can be correlated with amide II and amide III. In general, the shape of the obtained spectrum (Figure 2) correlates with the literature data [48,49,50]. Thus, the data obtained correlate with the literature data and the data of the European Directorate for the Quality of Medicines & HealthCare library [51], which confirms the assumption about the collagen nature of the resulting scaffold.

### 3.2. Fractional Composition Investigation of a Collagen Proteins Mixture

The fractional composition investigation was conducted in three steps: by size exclusion chromatography (Figure 3A), SDS-PAGE technique (Figure 3B,C), and MALDI-TOF (Figure 4). From the results of chromatography fractionation, we observed the presence of nine fractions in total with different migration rates. At the same time, more than 90% of the area of all peaks was accounted for by three fractions (Figure 3A, A/1, A/2, A/3). The elution rate of the most significant fractions corresponds to 0.4, 0.51, and 0.75–0.78 column volumes (CV). Typically, these elution rates correspond to collagen types I, II, and III.

The mixture of collagens isolated from the mantle of squid (*Dosidicus gigas*) was investigated in comparison to collagens from *Bos taurus* after collagenase treatment and trypsinolysis by SDS-PAGE (Figure 3B,C). The presence of proteins with molecular weight higher than 120 kDa was characteristic of the *Dosidicus gigas* collagen (Figure 3B, line N2), while the cattle (*Bos taurus*) collagen sample was enriched with the lower-molecular-weight proteins (Figure 3B, line N1). Next, we compared collagens from the different sources on SDS-PAGE after treatment with enzymes, collagenase (Figure 3B), and trypsin (Figure 3C) for 10, 30, and 30 min of reaction, respectively. Collagen from cattle demonstrates less sensitivity to collagenase as compared to collagen from hydrobionts (Figure 3B,C). Collagen from *Dosidicus gigas* likely has less sensitivity (higher resistivity) to trypsin, as the presence of higher-molecular-weight peptides after enzyme treatment was detected compared to *Bos taurus* collagen (Figure 3C, lanes 1 and 2, respectively).

The amino acid compositions were investigated by HPLC (Table 1), and the sequence of products of hydrolysis in the presence of collagenase and trypsin was analyzed by MALDI-TOF (Figure 4). The results of determining the amino acid sequence by the Top-Dawn MALD-TOF method (Figure 4) made it possible to determine the belonging of the isolated collagen fractions to a certain type. It was found that the amino acid composition of *Dosidicus gigas* collagen is similar to mammalian collagen (Table 1), but given the conservatism of its protein family, this is not surprising. This is also consistent with the notion that collagens are poor in aromatic amino acids. Thus, the data obtained support the assumption that the resulting protein and material from it can be attributed to collagens.

Molecular profiles of peptides after proteolytic digestion of collagens from *Dosidicus gigas* and *Bos taurus* by two proteases using MALDI-TOF mass spectroscopy were investigated. Collagen proteins from *Dosidicus gigas* (Figure 3A,B) form more complex profiles of maximums corresponding to proteolytic peptides appearing upon splitting by any of the two proteases than does collagen from cattle (compare Figure 3B,C, Table 1). Treatment of the *Dosidicus gigas* collagen solution with collagenase gives peptides with molecular weights of 481, 916, 1493, 2231, 2554, and 3808 Da. The same treatment of cattle collagen solution gives a pattern of peaks corresponding to 567, 916, 2666, 4148, and 5119 Da peptides. Mass spectroscopic analysis of trypsin splitting products of the two collagens demonstrates differences in the sensitivity of the two collagens to trypsin. There were 481, 1654, 2231, 2554, and 3808 Da peptides detected in the hydrobionts collagen solution, and 567, 1042, 2599, and 5422 Da peptides in the mammalian collagen solution. There were 481, 2231, 2554, and 3808 Da peptides detected in the *Dosidicus gigas* collagen solution after treatment with two types of proteases, which could be used as a representative. No similar pattern was detected for the cattle collagen solution.

Amino acid sequences of collagen fractions isolated from *Dosidicus gigas* and *Bos taurus* are given in Table 1. Peptides characteristic of collagen types I, II, and III were identified (Table 2). Interestingly, the products of collagen hydrolysis are different when treated with trypsin and collagenase. It can be assumed that in the case of the hydrobionts collagen, the amino acid sequence is Gly-Pro-Gly-Hyp-Gly-Pro-Hyp-Gly-Gly-Lys (938.99 Da), which is formed by two fragments with a molecular weight of ~481 Da. The masses of the peptides are predominantly determined by the frequency of lysine or arginine residue occurrence [51], which means that the more lysine or arginine residues in the collagen molecule, the more short-chain peptides will be observed in the hydrolysis products. It was found that α-chains are formed by a repeating sequence which can be written as follows: (Gly-XY)n. The identification of the amino acid sequence of the isolated peptides was carried out using FlexAnalysis software. Basic amino acids such as proline, glycine, and lysine were identified automatically by the program. On the assumption that the hydroxylysine molecule has a monoisotopic mass of 144.103, it was identified from this mass. Moreover, hydroxyproline has a monoisotopic mass that is similar to the monoisotopic mass of leucine/isoleucine, 113.04–113.08 , and it was assumed that the amino acids with a mass of 113.061 to 114.083 would correspond to hydroxyproline. Moreover, the mass of valine is extremely close to the mass of acetylated glycine, but since glycine is the most likely amino acid in collagen, it was decided that a mass of 99.032 to 99.05 would correspond to acetylated glycine. 

### 3.3. Visualization of Cell Compartments, Extracellular Matrix, and Estimation of Biocompatibility with Collagen Materials

As a result of the study of the suspension of HEK cells dyed by Mitotracker Red, DAPI, and FITC grown on the surface of the collagen scaffold, we observed the absence of cytotoxicity (Figure 5A). In the process of growth, cells probably actively use collagenous material as a source of amino acids, because they actively form intercellular contacts and a matrix. The obtained collagen scaffolds showed high cell viability, which suggests the possibility of biocompatibility of the resulting scaffold. A high cell survival rate was observed during the cultivation process (Figure 5B). Thereby, collagen matrix participates in the formation of a network of contacts and directs the process of tissue differentiation and participates in the formation of spatial morphology. In the MTT test, we found no toxicity from the collagen matrix, and cell survival was over 90%. The contribution of the collagen suspension was 3% in the pipetting medium.

### 3.4. High-Resolution X-ray Radiography on the Surface of the Collagen Matrix

High-resolution X-ray radiography on the laboratory complex “SynchrotronLike” made it possible to visualize the areas of the intercellular matrix forming on the surface of the collagen scaffold (Figure 6). The overlay of X-ray radiography and microscopy images of the area of the matrix with the nuclei of living cells stained with the DAPI dye made it possible to detect areas of the forming microenvironment of the connective tissue. The image shows separate areas of high density of the substance, which are associated with the formation of the extracellular matrix and the resulting living tissue. The method of high-resolution X-ray radiography was able to detect areas of high density coinciding with the structures of the extracellular matrix. Thus, this method can be applied to study the process of formation of the basal membrane and intercellular matrix on the surface of collagen matrices and to analyze their internal microstructure in laboratory conditions. The data obtained form the basis for the development of an optical scheme of experiments using coherent radiation from specialized sources of synchrotron radiation by the method of X-ray microtomography with a high time resolution.

## 4. Discussion

Collagen isolated from the mantle of the squid species *Dosidicus gigas* was used to create scaffolds for the cultivation of living cells. The protein fraction contains proteins that may correspond to collagen types I, II, and III. The technology for obtaining scaffolds based on collagen makes it possible to obtain this material on an industrial scale. In the process of forming scaffolds and thin films based on collagen, the self-structuring of fibrils occurs, while maintaining their tertiary structure. The topography of the collagen scaffold surface showed the presence of a developed rough surface, which ensures high cell adhesion. The mechanical properties of the obtained material make it possible to use it in surgical practice as a likely bioresorbable barrier material with a given sorption time and controlled biodegradation. The scaffolds obtained are not cytotoxic and have shown high biocompatibility. Direct visualization of the process of cell proliferation on the surface of the collagen scaffold using X-ray imaging made it possible to carry out remodeling of the spatial structure of the extracellular matrix, which is planned to be used in the future in the technology of printing 3D scaffolds to obtain living tissue with a given morphology and structure. The results of confocal microscopy indicate that the cell culture was growing on the surface of the studied collagen scaffold. Most likely, this can be explained by the high density of scaffold fibrils, which do not allow cells to penetrate inside. For 3D tissue prototyping, the obtained collagen should be reconstructed with the addition of a composite material (for example, polylysine). At the same time, the direction of surface cell growth is determined by the direction of the fibrils and can be used to provide controlled growth of connective tissue. The current results encourage the use of the resulting material as-is for a barrier scaffold. The resulting collagen scaffolds manifest themselves as a natural source of proteinogenic amino acids and provide a high rate of formation of the extracellular matrix.

## Figures and Tables

**Figure 1 polymers-15-01220-f001:** Mechanical and topography investigation results. (**I**) Measurement of the mechanical properties of the collagen scaffold: thickness, tensile strength, elongation ability, and Young’s modulus (**A**,**B**), and Young’s modulus result calculation (**C**). (**II**) Results of AFM measurements of collagen scaffolds and results of D-period calculation. (**III**) Results of SEM measurements of collagen scaffolds and results of analysis of developed pore structure and cell size: 1,2 (Panel **III**)—small-sized pores (5–12 mkm), 3,4 (Panel **III**) —large-sized pores (12–25 mkm).

**Figure 2 polymers-15-01220-f002:** IR spectroscopy of collagen scaffold obtained from *Dosidicus gigas* Pink curve is the *Dosidicus gigas* collagen scaffold spectrum.

**Figure 3 polymers-15-01220-f003:** Results for hydrobionts collagen fractionation by SDS-PAGE in different conditions. (**A**) Size exclusion chromatography of hydrobionts collagen. (**B**) Results of hydrobionts collagen type I, II, and III fractionation by SDS-PAGE after collagenase treatment. (**C**) Results of hydrobionts collagen type I, II, and III fractionation by SDS-PAGE after trypsinolysis.

**Figure 4 polymers-15-01220-f004:** MALDI-TOF mass spectrogram of *Dosidicus gigas* collagen (**A**,**B**) and *Bos taurus* collagen (**C**,**D**) proteolytic digestion products by (**A**,**C**) collagenase and (**B**,**D**) trypsin. [a.u.]—absorption units.

**Figure 5 polymers-15-01220-f005:** Results of confocal microscopy of collagen scaffold from *Dosidicus gigas* treated by cells stained by Mitotracker Red, DAPI, and FITC. (**A**) Fibroblasts, (**B**) viability estimation of cells. Mitochondria are stained in red; cell nuclei are stained in blue; matrix is stained in green.

**Figure 6 polymers-15-01220-f006:** Radiographic image of a collagen matrix for a sample of proliferating cells with an overlaid optical image of a region of the matrix with stained nuclei of living cells.

**Table 1 polymers-15-01220-t001:** Amino acid content in collagen from *Dosidicus gigas*.

Amino Acid	%	Amino Acid	%
Hydroxyproline (OHPro)	10.13	Methionine (Met)	1.39
Asparagines (Asp)	7.49	Isoleucine (Ile)	1.64
Threonine (Thr)	2.97	Leucine (Leu)	3.47
Serine (Ser)	3.86	Threonine (Tyr)	1.04
Glutamine (Glu)	11.38	Phenylalanine (Phe)	1.64
Proline (Pro)	9.40	Hydroxy lysine (OH-L)	1.50
Glycine (Gly)	22.18	lysine (Lys)	1.83
Alanine (Ala)	6.87	histidine (His)	1.07
Cysteine (Cys)	0.74	arginine (Arg)	8.79
Valine (Val)	2.61		

**Table 2 polymers-15-01220-t002:** Amino acid sequences of collagen fractions isolated from *Dosidicus gigas*.

Collagen Type
Collagen I	Collagen II	Collagen III
P*G	P*GPP*P*GE	P*GG*
KG	P*GI/LGE	KG
APTGGTTA	TAPP*	GHI/L
GPAG*AKDG*GYK*	P*HDP	I/LGCI/L
P*GDK*	P*P*EP*VG	CAG*I/L
PGMK*	P*P*EP*G*GGE	SMKG*PG
DK*I/LK*G*GG	P*P*EP*G*GEGM	SMMG*PG
	peptide - GVG P* M C PI/L	G*GP*G*
		P*G*GCK*

## Data Availability

Not applicable.

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
