# Peer review of "RETRACTED: The Dosidicus gigas Collagen for Scaffold Preparation and Cell Cultivation: Mechanical and Physicochemical Properties, Morphology, Composition and Cell Viability"

_polymers, 2023, doi:10.3390/polym15051220_

Round 1
Reviewer 1 Report
It can be accepted.
Author Response
We thank the reviewer for their attention to our manuscript. We have checked the quality of the text and tried to improve the quality of the English language.
Reviewer 2 Report
The author makes it difficult to check the changes that have been made. Where there is a substantial change, just saying "agreed, corrected" is not helpful, but rather it would be more helpful to include the corrected text in the response to reviewers letter. When I check to see what has been "corrected" I find in some instances it is not clear what has been done or why. A better response is needed before I can complete my review.
Author Response
We express our deep gratitude to the reviewer for the attention paid to our work. Our manuscript was prepared according to the standards of the Polymers journal. It consists of Introduction, Materials and Methods, Results and Discussion. In the introduction, we tried to formulate the research problem, in the Materials and Methods - described the techniques used in research. In the Results section, the stages of the work are sequentially described, and in the Discussion section, an attempt to generalize the data obtained was made. In general, the study includes the following steps.
- Study of the mechanical properties of scaffolds, it surface and composition. Young's modulus was evaluated. The surface was studied using AFM and SEM. The composition of the scaffolds was studied using FTIR. Protein fractions were investigated chromatographically and electrophoretically. An assessment was also made of the resistance of the obtained collagens proteins to various types of proteases, including collagenase from clostridia. The peptides obtained during hydrolysis were studied by MALDI-TOF. The amino acid composition of the proteins present in the sample was also investigated using HPLC.
- Visualization of cellular compartments, extracellular matrix and assessment of it possible biocompatibility with collagen materials was carried out using confocal microscopy and high-resolution radiography of the surface of the collagen matrix.
The following changes have been made in manuscript
- The title were corrected.
Original title
“Collagen from Dosidicus gigas for scaffold preparation and cell culture matrix analysis by
synchrotron irradiation”
Corrected title
“The Dosidicus gigas collagen for scaffold preparation and cell cultivation: mechanical and physicochemical properties, morphology, composition and cell viability.”
- The information about drying conditions and effect of drying were added and highlights by yellow in a section 2.2. Preparation Collagen Scaffold.
Of course, the wavelength of 380 nm does not cause sterilization. In this case, a technical error has been made. Sterilization at a wavelength of 280 nm was used in this work. Corrected and highlights by yellow in a section 2.2. Preparation Collagen Scaffold.
Original text:
“The technology for obtaining collagen material was developed. Collagen weighing 10 grams was suspended in 50 ml of a 3% glycerol solution in distilled water under ultrasound treatment. The suspension was placed on a Teflon substrate and dried at 40°C in a chamber under constant UV irradiation at 380 nm for cold sterilization. The resulting collagen sheets were used to obtain ready-to-use cell scaffolds.”
Corrected text:
“The technology for obtaining collagen material was developed. Collagen weighing 10 grams was suspended in 50 ml of a 3% glycerol solution in distilled water under ultrasound treatment. The suspension was placed on a Teflon substrate and airdried at 120°C in a convection dryer chamber until the moisture content in the sample is less than 3% under constant. Moisture content was controlled by the standard gravimetric method. The sample was dried to constant weight and the mass fraction of moisture was calculated as the difference in the masses of the sample before and after drying. The measurements were carried out in two repeats. Additionally, the moisture content was monitored using IR spectroscopy. After scaffold drying it was sterilized in UV at a wave-length of 280 nm and a temperature of 40°C. The resulting collagen sheets were used to obtain ready-to-use cell scaffold`s.”
- Expanded section 2.4. Atomic Force Microscopy. Extended information about instruments used for measurements
Original text:
“The study of the topography of the fibrillar structure and surface of collagen matrices was carried out by analogy with [35]. AFM imaging and indentation of collagen fibrils were performed using a Nanowizard AFM (JPK Instruments, Berlin, Germany). All measurements were taken in air and at room temperature. Aluminum-coated silicon AFM cantilevers with 150 kHz resonance frequency and 4.5 N/m nominal spring constant (NSC12 tip-C; MikroMasch, Tallinn, Estonia) were used. This cantilever was chosen to match the stiffness of collagen for optimizing sensitivity and signal/noise ratio (SNR). For imaging collagen fibrils without damage, the relatively high stiffness of the cantilevers required use of intermittent contact mode (also known as tapping mode). After taking a topographic image of the fibril, the force mapping mode (also known as force volume mode) was used to perform quasi-static indentations. The 64 × 64 indentation curves, consisting of 256 data points each, were taken on a square area. The area’s dimensions were 1.5 × 1.5 mm ± 0.5 mm, optimized in each experiment to balance approximately the number of data points taken on the fibril and on the substrate. The time for both the approach and retraction of the cantilevers was set to 0.2 s (5 Hz), with zero delay in-between.”
Corrected text:
“The study of the topography of the fibrillar structure and surface of collagen matrices was carried out by analogy with [35]. AFM imaging and indentation of collagen fibrils were performed using a Nanowizard AFM (JPK Instruments, Berlin, Germany) and SOLVER P47 (NT-MDT, Russia). All measurements were taken in air at room temperature. Aluminum-coated silicon AFM cantilevers (resonance frequency up to 150 kHz and nominal spring constant up to 4.5 N/m (NSC12 tip-C, MikroMasch, Tallinn, Estonia)) and gold-coated silicon AFM cantilevers (resonance frequency up to 150 kHz and nominal spring constant up to 6.1 N/m (NSG 03, NT-MDT, Moscow, Russia)) were used. These cantilevers were chosen to match the stiffness of collagen for optimizing sensitivity and signal/noise ratio (SNR). For imaging collagen fibrils without damage, the relatively high stiffness of the cantilevers required use of intermittent contact mode (also known as tapping mode). After taking a topographic image of the fibril, the force mapping mode (also known as force volume mode) was used to perform quasi-static indentations. The 64 × 64 indentation curves, consisting of 256 data points each, were taken on a square area. The area’s dimensions were 1.5 × 1.5 mm ± 0.5 mm, optimized in each experiment to balance approximately the number of data points taken on the fibril and on the substrate. The time for both the approach and retraction of the cantilevers was set to 0.2 s (5 Hz), with zero delay in-between.”
- Expanded section 2.5. Scanning Electron Microscopy. Added information about measurement modes.
Original text:
“Collagen scaffolds inclusive fixed overnight at 4°C with 3% glutaraldehyde in 0.05 M phosphate buffer (PB) at pH 7.2. Following extensive rinsing with the same buffer at 4°C, blocks were soaked for 1 h in 0.5% tannic acid in PB at 4°C. Then they were rinsed four times in the same buffer for 15 min at 4°C and post fixed with 1% osmium tetroxide in PB for 1 h at 4°C. Samples were washed in distilled water, dehydrated in a graded series of ethanol, and freeze-dried in a Balzers Union CPD 020 (Balzers, Liechtenstein) using the procedure of critical point drying. Samples were eventually sputter coated with gold in a Balzers MED 010 unit and observed in a JEOL JSM 6010LA electron microscope (Japan).”
Corrected text:
“Collagen scaffolds` inclusive fixed overnight at 4°C with 3% glutaraldehyde in 0.05 M phosphate buffer (PB) at pH 7.2. Following extensive rinsing with the same buffer at 4°C, blocks were soaked for 1 h in 0.5% tannic acid in PB at 4°C. Then they were rinsed four times in the same buffer for 15 min at 4°C and post fixed with 1% osmium tetroxide in PB for 1 h at 4°C. Samples were washed in distilled water, dehydrated in a graded series of ethanol, and freeze-dried in a Balzers Union CPD 020 (Balzers, Liechtenstein) using the procedure of critical point drying. Images of collagen scaffolds were obtained with scanning electron microscopy (SEM) in the bright field mode by the Jeol JSM-6390 (Japan) with accelerating voltageof 5 kV (low vacuum mode). As a SEM demands conductive surface to obtain an image, samples were covered by a platinum thin film (calibrated 30 nm) sputtered by the magnetron setup Jeol JFC-1600 (Japan).”
- Added section 2.9 FTIR spectra registration.
“FTIR spectra were obtained using a FT-IR ATR Alpha spectrometer equipped with a attenuated total reflectance unit (Bruker, Germany) and a InGAs detector. For FTIR spectra acquisition, the system was used in point mode (aperture of 250 × 250 μm) with a 2.0 cm−1 resolution and using 30 scans in transmittance mode (n = 35 spectra), and spectra acquisitions were performed under complete N2 purge of the analytical system.”
6. Expanded section 2.12. MTT assay for toxic effect determination. Added information about the shape of the scaffold, its preparation and the method of applying cells to it.
Original text:
“The effects of the collagen matrix on cell viability were determined using the MTT colorimetric test. HEK cells were diluted with the growth medium containing 100μМ CoCl2 for chemically induced hypoxia to 3.5 × 104 cells per ml and the aliquots (7 × 103cells per 200 μL) were placed in individual wells in 96-multiplates (Eppendorf, Germany) and incubated for 24 h. The next day the cells were then treated with synthesized compounds separately at 100 μМ concentration and incubated for 72 h at 37 °C in 5% CO2 atmosphere. Each compound was tested in triplicate. After incubation, the cells were then treated with 40 μL MTTsolution (3-(4,5-dimethylthiazol-2-yl)-2,5-diphenyltetrazolium bromide, 5 mg ml-1 in PBS) and incubated 4 h. After an additional 4 h incubation, the medium with MTT was removed and DMSO (150 μl) was added to dissolve the crystals formazan. The plates were shaken for 10 min. The optical density of each well was determined at 560 nm using a microplate reader ClarioStar (BMG Labteh, Germany). The tested collagen matrix was evaluated for the antiproliferative action in three separate experiments.”
Corrected text:
“The effects of the collagen matrix on cell viability were determined using the MTT colorimetric test. For the MTT test, collagen scaffold disks 30 mm in diameter and 60 µm thick were used. The MTT protocol has been optimized to increase well volume from 200 µl to 2000 µl. The discs were cut from edge to center once to form a 5 mm high cone. The cone was placed in a culture plate with cells 60 mm in diameter. The culture medium containing the cells was added in a volume of 2 ml (the cones were completely submerged). Cultivation was carried out in an incubator in a CO2 atmosphere. The culture plates were placed on an orbital shaker to simulate the movement of the culture fluid. HEK cells were diluted with the growth medium containing 100μМ CoCl2 for chemically induced hypoxia to 3.5 × 104 cells per mL and the aliquots (7 × 103 cells per 200 μL) were placed in individual wells in 96-multiplates (Eppendorf, Germany) and incubated for 24 h. The next day the cells were then treated with synthesized compoundsseparately at 100 μМ concentration and incubated for 72 h at 37 °C in 5% CO2 atmosphere. Each compound was tested in triplicate. After incubation, the cells were then treated with 40 μL MTT solution (3-(4,5-dimethylthiazol-2-yl)-2,5-diphenyltetrazolium bromide, 5 mg ml-1 in PBS) and incubated 4 h. After an additional 4 h incubation, the medium with MTT was removed and DMSO (150 μl) was added to dissolve the crystals formazan. The plates were shaken for 10 min. The optical density of each well was determined at 560 nm using a microplate reader ClarioStar (BMG Labteh, Germany). The tested collagen matrix was evaluated for the antiproliferative action in three separate experiments.”
- In Figure 1 were added results of AFM with 2 resolutions 5 mkm and 3 mkm.
- The results of the investigation using FTIR (Fig. 2) and their analysis are added in manuscript.
“According to the literature collagens type I, III, IV, and V have several characteristic FTIR regions located in the ranges of 1.700–1.600 cm–1, 1.480–1.350 cm–1, 1.300–1.180 cm–1, and 1.100–1.005 cm–1 (47). In addition, there is evidence that FTIR spectra of fish skin collagens demonstrate the presence of amides (3423 and 3337 cm-1 for amide A and 2928 and 2924 cm-1 for amide B, respectively). In addition, bands correspond to stretching vibrations of C-O of the polypeptide backbone in the region of 1600-1700 cm-1, hydrogen bonding between the N-H and C-O (Gly) residues at 1654 and 1560 cm-1. Signals for amide II and amide III in structure of collagen were detected at the 1240 cm-1 (48). At the same time, FTIR spectra obtained for collagen extracted from the skin of squid (Doryteuthis singhalensis) have similar characteristics (48).
FTIR investigations of the obtained scaffold were carried out (Fig. 2). The obtained data in-dicate the presence of bands in the region of 3300 cm-1 and 2935 cm-1, which may correspond to amide A and amide B, a band in the region of 1659 cm-1, which can be correlated with the C-O backbone of the polypeptide chain, a band in the region of 1538 cm-1, which is consistent with regions characteristic of N-H and C-N. A band was also detected, in the region of 1235 cm-1, which can be correlated with amide II and amide III. In general, the shape of the obtained spectrum (Fig. 2) correlates with the literature data (47, 48, 49, 50, 51). Thus, the data obtained correlate with the literature data and the data of the European Directorate for the Quality of Medicines & HealthCare library (https://www.edqm.eu), which confirms the assumption about the collagen nature of the resulting scaffold.”
- Information about the amino acid composition of collagen was taken out of Figure 3D into a separate table and an analysis of the results was added.
“It was found that the amino acid composition of Dosidicus gigas collagen is similar to mammalian collagen (Table 1), but given the conservatism of it proteins family it not surprising. This is also consistent with the notion that collagens are poor in aromatic amino acids. Thus, the data obtained support the assumption that the resulting protein and material from it can be attributed to collagens.”
- In section 4. Discussion discussed a reasons for the surface location of cells on the scaffold and the possibility of it use has been added.
“The results of confocal microscopy indicate that the cell culture is growing on the surface of the studied collagen scaffold. Most likely, this can be explained by the high density of scaffold fibrils, which do not allow cells to penetrate inside. For 3D tissue prototyping, the obtained collagen should be reconstructed with the addition of a composite material (for example, polylysine). At the same time, the direction of surface cell growth is determined by the direction of the fibrils and can be used to provide controlled growth of connective tissue. Current results are encouraging to use the resulting material as is for a barrier scaffold.”
- Correct Line 28
Original text “The resulting material provides the forming of the extracellular matrix and characterized by a limited time to living tissue sorption.”
Corrected text “The resulting material provides the forming of the extracellular matrix and characterized by a short time to living tissue sorption.”
12. In Line 78 and line 147 “to analogy” changed to “by analogy”
13. In Line 138 and 139 “Seize” changed to “size”.
14. Line 146, 147, 152 (and other places in the manuscript) PAAG and PAAGE changed to PAGE.
15. In Line 202 “Scaffolds” changed to “Scaffold’s”.
16. In Line 202 “were investigate by tree in dependent technic.” corrected to “were investigated by three independent technique.”
17. In Line 208 “It was detected that the pitch of the turns in the collagen molecules helix (D-period) is 67 nm” changed to “The results of collagen molecules surface topography investigation made possible to estimate the value of the D-period of it fibrils, which was 68 ± 1.8 nm for collagen proteins and correlate with literature [24].”
- Author contributions: Most of the work seems to have been done by XX, YY and ZZ but these authors are not listed in the author list.
Added information about authors contribution.
“Author Contributions: Conceptualization, E.C. and V.A.; methodology, M.K., K.M. and D.B.; software, A.N.; validation, A.B., S.A., and L.A.; formal analysis, V.A.; investigation, V.A., M.G., D.B., and O.A.; resources, O.B. and A.G.; data curation, S.A.; writing—original draft preparation, E.C.; writing—review and editing, E.C.; visualization, S.A.; supervision, S.A.; project administration, A.S.; funding acquisition, S.A. All authors have read and agreed to the published version of the manuscript.”

Reviewer 3 Report
This manuscript reports the extraction of collagen from Dosidicus gigas, used to develop 3D scaffolds for tissue engineering applications. This is an interesting paper and can be accepted for publication after the following amendments:
· Line 102-104. "The suspension was placed on a Teflon substrate and air dried at 40 °C in a convection dryer chamber until the moisture content in the sample was less than 3% under constant UV irradiation at 280 nm for soft cold sterilization." How did the authors control or measure the moisture content? Furthermore, is the sterilization at 40 °C a soft cold sterilization?
· What is the size of the scaffold used for the MTT assay?
· What is the size of collagen fiber and thickness of the scaffold? Discuss in detail.
· Separate Fig. 2D from Figure 2 and put it in a separate table.
· For a plausible conclusion, please measure the extract's FTIR spectrum.
· It would be reasonable to claim a 3D scaffold, if the author could show the cell infiltration through the scaffold.
Author Response
We thank to reviewer for valuable comments and recommendations. We made changes in the manuscript which can be found in the attachment. Please see the attachment.

Round 2
Reviewer 2 Report
Suitable for publication now